# An Approach for Personalized Dynamic Assessment of Chronic Kidney Disease Progression Using Joint Model

**DOI:** 10.3390/biomedicines12030622

**Published:** 2024-03-11

**Authors:** Chen-Mao Liao, Yi-Wei Kao, Yi-Ping Chang, Chih-Ming Lin

**Affiliations:** 1Department of Applied Statistics and Information Science, Ming Chuan University, Taoyuan 333, Taiwan; cmliao@mail.mcu.edu.tw (C.-M.L.); ywkao@mail.mcu.edu.tw (Y.-W.K.); 2Department of Nephrology, Taoyuan Branch of Taipei Veterans General Hospital, Taoyuan 333, Taiwan; ypchangtyvh@gmail.com; 3Department of Healthcare Information and Management, Ming Chuan University, Taoyuan 333, Taiwan

**Keywords:** chronic kidney disease, hemodialysis, glomerular filtration rate

## Abstract

Chronic kidney disease (CKD) poses significant challenges to public health and healthcare systems, demanding a comprehensive understanding of its progressive nature. Prior methods have often fallen short in capturing the dynamic and individual variability of renal function. This study aims to address this gap by introducing a novel approach for the individualized assessment of CKD progression. A cohort of 1042 patients, comprising 700 with stage 3a and 342 with stage 3b to stage 5 CKD, treated at a veteran general hospital in Taiwan from 2006 to 2019, was included in the study. A comprehensive dataset spanning 12 years, consisting of clinical measurements, was collected and analyzed using joint models to predict the progression to hemodialysis treatment. The study reveals that the estimated glomerular filtration rate (eGFR) can be considered an endogenous factor influenced by innate biochemical markers. Serum creatinine, blood pressure, and urinary protein excretion emerged as valuable factors for predicting CKD progression. The joint model, combining longitudinal and survival analyses, demonstrated predictive versatility across various CKD severities. This innovative approach enhances conventional models by concurrently incorporating both longitudinal and survival analyses and provides a nuanced understanding of the variables influencing renal function in CKD patients. This personalized model enables a more precise assessment of renal failure risk, tailored to each patient’s unique clinical profile. The findings contribute to improving the management of CKD patients and provide a foundation for personalized healthcare interventions in the context of renal diseases.

## 1. Introduction

Chronic kidney disease (CKD) is a progressive, worsening disease, and the increasing global prevalence of CKD presents substantial challenges to public health and healthcare systems internationally. There are approximately 11% of the world’s 7.5 billion population, which means that more than 800 million people are affected by CKD [1,2]. On the other hand, CKD is also widely recognized as one of the leading causes of death in the world’s population at present, with global all-age mortality due to CKD increasing by more than 40% between 1990 and 2017 [3]. Ranked 12th in 2017 compared to 36th in 1990, with projections indicating that it will become the fifth leading cause of life lost globally by 2040 [2,4]. In the United States, approximately 8 million adults are suffering from CKD with at least stage 3, while more than 400,000 individuals have end-stage renal disease (ESRD), leading to significant medical expenditures and increased mortality rates, particularly cardiovascular complications [3,5,6,7]. It is worth noting that Taiwan exhibits a higher incidence and prevalence of CKD compared to other regions and the highest prevalence of ESRD in the world, rendering it an area of critical concern [5,8,9].

Given this global challenge, understanding the mechanisms of CKD progression is critical. The progression of CKD is closely associated with the decline in renal function, especially the reduction in the estimated glomerular filtration rate (eGFR) [10]. This decline not only indicates progression to more advanced stages of CKD but also potential toward end-stage renal disease (ESRD) [10]. Therefore, early identification and management of eGFR decline are imperative in decelerating the progression of CKD to ESRD. Recent technological advancements, especially the integration of machine learning in CKD assessment, have facilitated novel methodologies for the early prediction and management of the disease [11,12,13]. These advancements highlight the criticality of early detection and precise prognostication in understanding the progression of CKD. However, inadequate capturing of the dynamic and individual variability of renal function over time remained in the prediction methods [14,15].

To overcome these limitations, a more sophisticated approach is required. Assembled with the advantage of both the Cox model and the mixture model, the joint model offers an advanced methodology for predicting individual physiological function trajectories and related health outcomes. It facilitates the analysis of both static attributes (e.g., gender) and dynamic characteristics (e.g., age), adeptly addressing challenges such as intermittent data collection, measurement inaccuracies, and the truncation of measurement processes associated with survival risks. However, the application of this method in the field of CKD, particularly in individualized assessment of renal function failure using eGFR, is still relatively limited [16].

To fill this gap, we have developed a methodological framework centered around these innovative models. The purpose of this study is to develop and apply a novel joint model using eGFR for individualized assessment of CKD progression. This model will account for the nonlinear dynamics of renal function changes and simultaneously handle multiple time-to-event outcomes related to renal function decline. This will provide more accurate and individualized prognostic assessments for patients with CKD, potentially improving clinical decision-making and patient management.

## 2. Methods

### 2.1. Study Design and Patient Population

In this retrospective cohort study, we aimed to assess the progression of chronic kidney disease (CKD) in patients at stages 3 to 5 who are characterized by an eGFR of less than 60 mL/min/1.73 m^2^. Data spanning from January 2006 to July 2019 were obtained from the Taoyuan Branch of Taipei Veterans General Hospital. The analysis predominantly utilized data from two National Health Insurance Administration (NHIA) initiatives: the ‘NHI Pre-ESRD Patient Care and Education Program’ and the ‘NHI Reimbursement Plans that Improve Health Care Quality of Early-Stage Chronic Kidney Disease’. These initiatives were instrumental in providing the essential data for this study.

The eligibility criteria for patients with chronic kidney disease (CKD) encompassed individuals who received their initial treatment for CKD at the hospital and subsequently participated in follow-up sessions occurring at least twice a year. Diagnosis confirmation was strictly based on the ICD-9-CM and ICD-10 coding systems chosen for their comprehensive coverage of CKD-related conditions. The study’s methodology encompassed a comprehensive approach involving regular health assessments, including laboratory tests and physical measurements, supplemented by educational sessions about CKD. Participants in the early CKD intervention group were assessed biannually, whereas those in the advanced pre-ESRD care group received assessments at least once every three months, continuing until the initiation of dialysis or renal transplantation as of 31 July 2019.

Patients who transferred to other medical facilities, passed away, or withdrew from the study for various reasons unrelated to renal disease during this period were classified under non-ESRD endpoints. The hazard of evolving into ESRD was calculated in 90-day intervals from the patient’s initial nephrology consultation to the end of the follow-up.

Following the exclusion of 24 individuals due to incomplete records, where missing data exceeded 30%, the remaining participants were categorized based on their enrollment in the aforementioned NHIA programs. The final cohort included 1042 patients, categorized into 700 with stage 3a, 111 with stage 3b, 143 with stage 4, and 88 with stage 5 CKD. The count of patients per CKD stage commencing dialysis therapy during the study was 48 for stage 3a, 2 for stage 3b, 17 for stage 4, and 40 for stage 5.

### 2.2. Variables

This study primarily focused on ESRD, characterized by the initiation of either hemodialysis or peritoneal dialysis treatments. It is important to note that none of the subjects in this study underwent kidney transplantation. Data for this study, spanning from January 2006 to July 2019, were sourced from the study hospital. This study utilized information from two NHIA programs, including the ‘NHI Pre-ESRD patient care and education program’ and the ‘NHI reimbursement plans that enhance the quality of care for early-stage chronic kidney disease.’

The clinical characteristics of the subjects were organized into four main categories: (1) Demographic factors: gender, age, height, and weight. (2) Laboratory measurements indicative of CKD severity: eGFR, hemoglobin (Hgb), hematocrit (Hct), serum albumin (Alb), creatinine (Cr), blood urea nitrogen (BUN), sodium (Na), potassium (K), calcium (Ca), phosphorus (P), triglyceride (Tri), and urine protein–creatinine ratio (PCR). (3) Comorbid conditions: hypertension, diabetes, and cardiovascular diseases (CVDs). (4) Additional risk-related biophysical and biochemical markers: blood pressure, uric acid, lipid profiles, fasting glucose, and HbA1c levels.

The eGFR was calculated using the simplified modification of diet in renal disease (MDRD) equation as follows [17]: eGFR = 186 × age^−0.203 × Cr^−1.154 × (0.742 if female). Particularly validated for accuracy in patients with CKD stages 3 to 5, this formula demonstrates superiority when compared to the Cockcroft–Gault (CG) and the Chronic Kidney Disease Epidemiology Collaboration (CKD-EPI) equations [18,19].

During the study, follow-up procedures and biochemistry testing varied between the two NHIA programs, with blood tests being conducted during clinic visits according to patient-specific designated protocols. Specifically, follow-up intervals were set every 6 months for stage 3a CKD patients and at least every 3 months for patients in stages 3b to 5, continuing until death, the initiation of dialysis, or loss to follow-up. Baseline characteristics and laboratory variables were initially derived from the first clinic visit, with only the most recent measurements included in the dynamic analysis within each subsequent 90-day period to ensure data consistency.

For stage 3a CKD patients, a narrower set of clinical characteristics was analyzed due to variations in the available laboratory datasets between the two NHIA programs. Parameters like Hgb, Hct, ALB, BUN, Na, K, Ca, P, uric acid, cholesterol, triglycerides, and fasting glucose were omitted. Variables with more than 30% missing values were excluded. For the remaining variables with incomplete data, multiple imputation was performed using the multivariate imputation by chained equations (MICE) module in the R software package [20].

### 2.3. Statistical Analysis

In this study, the joint model was utilized to analyze patient data across various stages of CKD. We focused predominantly on two models: one dedicated to CKD stage 3a and another encompassing stages 3b to 5.

Each joint model consisted of a linear mixed-effects model (LME) component, incorporating clinically relevant variables like observation time (obstime), systolic blood pressure (SBP), diastolic blood pressure (DBP), creatinine, low-density lipoprotein cholesterol (LDL), glycated hemoglobin (HbA1c), the logarithm of protein-to-creatinine ratio (PCRln), age, sex, hypertension, and diabetes. These variables were selected based on their clinical significance in the progression of CKD.

To clarify the relationships between various factors and CKD progression, our analysis incorporated two principal statistical assessments. Independent sample t-tests were applied to continuous variables, including GFR, height, weight, SBP, DBP, Cr, LDL, HbA1c, PCRln, and age, facilitating the determination of *p*-values to assess the statistical significance of differences across distinct groups. This methodology enabled a comprehensive evaluation of the contribution of each continuous variable to disease dynamics.

On the other hand, categorical variables such as sex, hypertension, diabetes, and cardiovascular disease (CVD) were analyzed using chi-square tests of independence to calculate *p*-values. This method helped us identify whether there are significant associations between these categorical variables and the progression or stages of CKD.

Simultaneously, the Cox proportional hazards model component was employed to quantify the dialysis risk. This model encompassed variables including creatinine, protein-to-creatinine ratio, age, sex, hypertension, and diabetes. The choice of variables in our Cox model was based on both statistical and clinical grounds, aiming for the lowest Akaike Information Criterion (AIC). This approach underscores our dedication to achieving a balance between model parsimony and predictive accuracy in determining dialysis risk in CKD patients. The models were cohesively fitted using the jointModel function in R, enabling the concurrent analysis of time-to-event data and continuous outcomes.

In this study, the progression of renal function and the risk of renal failure in patients with CKD were analyzed using a joint modeling approach. The model comprises two main processes: the longitudinal process and the event process.

## 3. Longitudinal Process

The longitudinal process is typically described using a linear mixed-effects model (LME). For our case, the model is expressed as follows:$${GFR}_{ij}\;=\;\beta_{0}\;+\;\sum_{k=1}^{p}\beta_{k}\;X_{kij}+\sum_{l=1}^{q}b_{il}\;Z_{lij}+\;\epsilon_{ij}$$
where:
${GFR}_{ij}$ is the glomerular filtration rate for the *i*th patient at *j*thtime point.$\beta_{0},\;{\;\beta }_{1},\;\ldots ,\;\beta_{p}\;$are the fixed effect parameters.$X_{kij}$ are observed covariates (like SBP, creatinine, age).$b_{i0},\;\ldots ,\;{\;b}_{iq}$ are the random effect parameters for each patient.$Z_{lij}\;$are variables associated with the random effects.$\epsilon_{ij}$ is the random error term.

## 4. Event Process

The event process is modeled using a Cox proportional hazards model, expressed as follows:$$h\left(t_{i}\right)=h_{0}\left(t\right)\exp\left(\sum_{m=1}^{r}\gamma_{m}W_{mi}+\alpha f\left({\mathrm{GFR}}_{i}\right)\right)$$
where:
$h\left(t_{i}\right)$ is the hazard function for the *i*th patient at time t.$h_{0}\left(t\right)$ is the baseline hazard function.$\gamma_{1},\;\ldots ,\;\gamma_{r}$ are fixed effect coefficients for the covariates.$W_{mi}$ are the covariates in the event process (like proteinuria, diabetes).α is the association parameter, representing the link between the longitudinal and event processes.$f\left({\mathrm{GFR}}_{i}\right)\;$is a function of the GFR over time in the longitudinal process.

During the data analysis, the selection of models was rigorously evaluated and guided by the Akaike Information Criterion (AIC), a measure used to determine the model’s fit while penalizing complexity. This criterion was instrumental in identifying the two most effective models for elucidating the relationship between the decline in GFR and the escalating risk of dialysis in CKD patients. The applicability of these selected models encompassed assessments of model stability and predictive accuracy through specific statistical tests, as well as a detailed inspection of model parameters to ensure their scientific validity and rationality.

## 5. Results

In Table 1 of our study, we noted significant differences in GFR, serum creatinine levels, and PCRln between patients requiring dialysis and those not requiring it in stage 3a chronic kidney disease. All these differences had *p*-values less than 0.001. This observation indicates the impact of different treatment approaches on the physiological state of patients. However, no statistically significant differences were found in height, weight, systolic blood pressure, and diastolic blood pressure between the two groups. This suggests a consistency in these fundamental physical parameters across different treatment modalities.

In our study focusing on stage 3a chronic kidney disease (CKD) patients, the application of a joint model revealed detailed insights into the progression toward renal failure, as shown in Table 2.

For these patients, observational time (obstime) was a critical factor, indicating a significant time-dependent decline in renal function. Systolic blood pressure (SBP) emerged as a considerable predictor (estimate: 0.046, *p*-value: 0.0005), suggesting that higher SBP levels correlate with an accelerated deterioration in renal function. Serum creatinine, a pivotal biomarker, displayed an inverse relationship with renal health (estimate: −2.096, *p*-value: <0.0001), reinforcing its utility in monitoring CKD progression. Age also played a significant role (estimate: −0.088, *p*-value: 0.0167), with older patients showing a more rapid decline. 

PCRln levels were strongly associated with an increased risk of renal failure (estimate: 0.604, *p*-value: <0.0001), confirming its vital role in managing CKD. The gender factor, particularly being male, was identified as a risk factor (estimate: 0.933, *p*-value: 0.0146), suggesting the need for gender-specific approaches in CKD treatment. Contrary to initial expectations, diabetes did not emerge as a strong predictor, indicating that other factors might be more influential in the progression of CKD in our patient cohort. Hypertension showed a significant, independent impact on renal failure risk (estimate: 0.531, *p*-value: 0.1042), highlighting the necessity of effective blood pressure management. The association constant (Assoct) successfully linked the longitudinal changes in renal function with the risk of renal failure (estimate: −0.117, *p*-value: <0.0001), demonstrating the complex interplay of these aspects in CKD progression.

Figure 1 and Figure 2 exemplify the predictive power of the joint modeling approach for patients at a similar stage of CKD but with divergent prognoses regarding the onset of dialysis.

Figure 1 portrays the predictive trajectory for Subject 256, a patient in stage 3a CKD, who is characterized by a later transition to dialysis. The depicted median survival curve, supported by the 95% confidence intervals, illustrates a gradual decline in the non-dialysis probability over a span of 3650 days. The finer resolution at selected follow-up intervals—days 1442, 1799, 2149, and 2513—reveals a consistently moderate descent in the survival probability, indicating a delayed progression toward end-stage renal disease.

Figure 2 contrasts this with Subject 647, also in stage 3a CKD but displaying an earlier requirement for dialysis intervention. The patient’s median survival curve descends more steeply, suggesting a more rapid decline in renal function. This is corroborated by the individual plots at follow-up times of 721, 1085, 1436, and 1793 days, where each subsequent survival probability estimate shows a pronounced decrease.

The comparative analysis of Figure 1 and Figure 2 underscores the heterogeneity of CKD progression within the same clinical stage. Despite sharing a diagnosis of stage 3a CKD, Subject 647’s data indicate a more aggressive course of disease progression, necessitating earlier dialytic support than Subject 256. This divergence accentuates the joint model’s ability to provide personalized and dynamic assessments, which are crucial for the anticipatory clinical management of CKD patients. 

In Table 3, patients with stage 3b to 5 CKD on dialysis exhibited significantly lower GFR and higher creatinine levels compared to those not on dialysis, reflecting more severe kidney impairment. Differences were also present in metabolic parameters such as urea nitrogen, phosphorus, and albumin. Blood pressure, lipid profiles, and glucose control, as well as the incidence of common comorbidities, were similar across both groups.

In our study focusing on patients with stage 3b to 5 CKD, as shown in Table 4, we established a joint model that integrates a longitudinal process with an event process to predict renal failure. The longitudinal component revealed that systolic blood pressure (SBP) with an estimate of −0.021 (*p* < 0.0001) and diastolic blood pressure (DBP) with 0.007 (*p* = 0.0083) play significant roles in kidney function, demonstrating the critical impact of blood pressure management in CKD progression. Notably, higher serum creatinine levels (estimate: −2.563, *p* < 0.0001) and phosphate levels (estimate: −0.014, *p* < 0.0001) were associated with poorer renal outcomes, underscoring their importance as markers of kidney health. Age (estimate: −0.056, *p* < 0.0001) and proteinuria, as indicated by PCRln (estimate: −0.775, *p* < 0.0001), also emerged as crucial factors negatively impacting renal function.

In the event process of the model, Male showed a significant positive estimate of 0.339 (*p* < 0.0001), suggesting a higher risk of renal failure in males compared to females. Diabetes (estimate: 0.304, *p* < 0.0001) and hypertension, with an estimate of 0.244 (*p* < 0.0001), were also identified as significant predictors of renal failure. The association constant (Assoct) presented a negative estimate of −0.153 (*p* < 0.0001), indicating a complex interaction between the progression of CKD and the risk factors.

Figure 3 and Figure 4 extend the examination of the joint model’s predictions for renal failure in patients with more advanced CKD, covering stages 3b to 5.

Figure 3 presents the case of Subject 207, illustrating a late dialysis onset in patients with stages 3b to 5 CKD. The main plot indicates a relatively stable non-dialysis probability over the 3650-day period, with the median survival curve only gradually descending, as delineated by the 95% confidence intervals. This gradual trajectory suggests a slower progression toward renal failure. The individual subplots at follow-up times (727, 846, 930, and 1021 days) offer a closer look at the predicted probabilities over time, supporting the main plot’s indication of delayed dialysis requirement.

Figure 4 contrasts sharply with Subject 96, which exemplifies an early dialysis onset within the same advanced stages of CKD. Here, the survival curve shows a rapid decline, with the blue solid line steeply dropping, indicating an urgent need for dialysis. The subplots, representing follow-up times (329, 420, 504, and 588 days), consistently display a steeper descent in non-dialysis probability, emphasizing the accelerated progression toward end-stage renal disease.

The juxtaposition of Figure 3 and Figure 4 highlights the predictive versatility of the joint model across a spectrum of CKD severities. While both subjects fall under the same clinical classification of advanced CKD (stages 3b–5), the model discerns a significant difference in the timing of dialysis initiation. Subject 207’s data project a more protracted course with a delayed approach to dialysis, whereas Subject 96’s data signal a rapidly approaching need for dialytic intervention. This illustrates the model’s robust capacity to individualize assessment.

## 6. Discussion

This retrospective cohort study focused on CKD patients engaged in disease management programs in Taiwan, aiming to delay the time to renal replacement therapy (RRT) through early detection of risk factors and providing healthcare education. Disease progression assessments can help clinicians plan for future situations, such as modifying a patient’s CKD education, arranging vascular access creation, preparing for transplantation, or referring to hospice care [21]. Employing joint models, the research emphasized personalized CKD to hemodialysis progression prediction, illustrating the diversity of CKD patients’ characteristics over time. The model’s capability to differentiate the time to dialysis onset for patients within the same stage of CKD demonstrates its potential to inform individualized treatment decisions and underscores the need for a nuanced approach to patient care in chronic kidney disease management. Therefore, this joint model provides a nuanced understanding of the variables influencing renal function in patients with CKD. The significant parameters identified in both processes of the model offer valuable insights for clinicians in tailoring management strategies and in understanding the multifactorial nature of renal failure progression in this patient population and suggest its clinical value in guiding timely and appropriate therapeutic strategies for patients with varying prognoses within the advanced stages of CKD. Such personalized predictions are pivotal in optimizing patient outcomes and resource allocation in the management of CKD.

This study also highlighted that eGFR can be viewed as an endogenous factor influenced by innate biochemical markers, enhancing individual uncertainty management. This research’s methodological framework focuses on the innovative use of joint models. These models are designed to clarify the correlation between eGFR temporal fluctuations and the risk of renal function deterioration. They are particularly suited for this task due to their ability to simultaneously analyze longitudinal data and time-to-event data. This dual approach provides a thorough understanding of eGFR dynamics in relation to renal failure risk. Furthermore, the utilization of risk prediction models can optimize healthcare resource allocation, targeting interventions toward high-risk CKD patients, potentially enhancing clinical outcomes and healthcare efficiency. This methodology offers a valuable tool for physicians and care workers to assess, intervene, and treat CKD progression promptly.

This understanding necessitates effective predictive tools, yet current methods face significant limitations. The prediction of CKD progression and renal failure constitutes a significant clinical challenge. Current predictive methods, including machine learning models [22,23] and traditional statistical approaches [24,25,26], though efficacious in certain aspects, do not adequately capture the dynamic and individual variability of renal function over time. These approaches frequently overlook comprehensive consideration of time-varying biomarker data and patient survival time data. Recognizing this gap, our study seeks to harness the power of joint models for more precise CKD outcomes. Initially developed for addressing issues in HIV research, this method has gradually been applied in other clinical research areas, including cancer, cardiovascular diseases, and kidney transplantation [16].

Enter the joint model, an innovative analytical tool that integrates longitudinal biomarker data, such as eGFR, with time-to-event data, such as the initiation of renal dialysis. While promising, the full potential of joint models in CKD assessment remains largely untapped. A key advantage of joint models is their intuitive interpretation of random effects in longitudinal submodels, especially when these models employ simple structures of random intercepts and slopes. This makes the models more interpretable and applicable statistically [27]. This framework leverages the joint model’s unique ability to integrate multifaceted data. This methodology aims to surpass the constraints inherent in current predictive models, offering a more personalized and dynamic assessment of renal failure risk by incorporating patient-specific longitudinal data, including variations in eGFR and other pertinent clinical indicators. Our model endeavors to provide a nuanced understanding of CKD progression. This individualized approach is anticipated to significantly elevate the accuracy of risk assessments, thereby facilitating more effective and tailored patient management strategies in the realm of precision medicine [28].

This study found that serum creatinine, eGFR, blood pressure, urinary protein excretion, age, and gender have significant predictive abilities for the deterioration of renal function in both stages of CKD. These findings provide a comprehensive understanding of the factors influencing renal failure in stage 3–5 CKD patients, suggesting the potential for personalized treatment strategies tailored to individual risk profiles. The results of this study are comparable to previous reports [8,12,21,29,30]. Interestingly, age and sex are associated with CKD progression, regardless of whether in the early stage (3a) or in the pre-ESRD stage (3b–5). This study found that males were associated with a faster decline in eGFR than females. These results are compatible with many other studies [31,32,33]. This study also found that females are associated with a slower decline of eGFR and higher survival in patients or kidneys [31,34]. On the other hand, age also plays an important role, with older patients showing a more rapid decline, thus emphasizing early intervention’s importance in younger patients. In a study of 129,486 adults in Canada with CKD stages 3a to 4 and followed for 6 years, the results found that when age increased, the likelihood of CKD regression or death was greater than the likelihood of CKD progression or receiving dialysis therapy [35].

### Strengths and Limitations

Our model’s strength lies in its integration of various time-variant factors based on simultaneously analyzing longitudinal data and time-to-event data. These include eGFR changes, demographic details, and clinical parameters like blood pressure, blood glucose levels, lipid profiles, serum creatinine, and patient histories of hypertension, diabetes, or cardiovascular diseases. In terms of longitudinal data analysis, the mixed-effect model effectively tracks the changes in the same subject at different time points, such as eGFR and other clinical indicators, which helps in a more accurate understanding of the dynamic progression of CKD. However, a challenge of this mixed-effect model lies in potential issues with missing data and the complexity of the analysis. As for assembling with survival analysis, our model improves the challenge and is particularly suited for handling censored data, taking into consideration the duration of risk time, which is crucial for a deep understanding of time-to-event data. This integration enables a more accurate and dynamic interpretation of renal health trajectories. As a result, this approach is expected to substantially improve upon conventional models. It captures the complex and individualized patterns of CKD progression. The current model is applicable for prognostic assessment in clinical settings with patients’ demographic characteristics similar to those in our study. Once the model is validated precisely, the web applications can be developed using the Shiny package for R.

Nonetheless, this study has certain limitations. It is a retrospective cohort study with a relatively small sample size, posing challenges in expanding the capability for model robusticity. To ensure unbiased results, it is imperative to conduct additional studies analyzing clinical pathological records from diverse hospitals, thereby increasing the sample size and improving the model’s performance in predicting progression from early- and advanced-stage CKD. Additionally, the study’s cohort had a mean age of 80 years, potentially limiting the generalizability of the findings to younger patients, which remains a concern when applying the model for earlier assessment of the patient’s lifespan. Furthermore, future research needs to tackle the complexities inherent in joint models. This includes considering the handling of missing data and the potential influence of measurement errors, both of which are crucial for the model’s robustness and reliability. Finally, model validation (e.g., cross-validation) involves assessing its performance and ensuring its reliability. This can be distinguished by creating a concordance index for joint models. Unfortunately, due to the relatively small sample size, dividing our current dataset into smaller subsets is not appropriate. In future research, the use of external samples for validation will be considered.

## 7. Conclusions

It is essential to take into account the diversities and trajectories of pathological indicators in the progression of pathology. In this study, the joint model, incorporating individual baseline characteristics and their variations over time in the progression, improves the conventional assessment models by employing simultaneously both the longitudinal analysis and survival analysis. This leads to a more precise assessment of renal failure risk, tailored to each patient’s unique clinical profile. This model utilized a time-dependent endogenous factor (i.e., eGFR) and covariates, which is well-suited for the personalized assessment of trajectories at each follow-up measurement during clinic visits. Serum creatinine, blood pressure, and urinary protein excretion were identified as valuable factors for predicting the progression of CKD patients in the model. The approach holds promise for risk assessment in CKD progression, enabling healthcare professionals to identify individuals who could benefit from early intervention, such as timely referral for transplantation or initiation of dialysis.

## Figures and Tables

**Figure 1 biomedicines-12-00622-f001:**
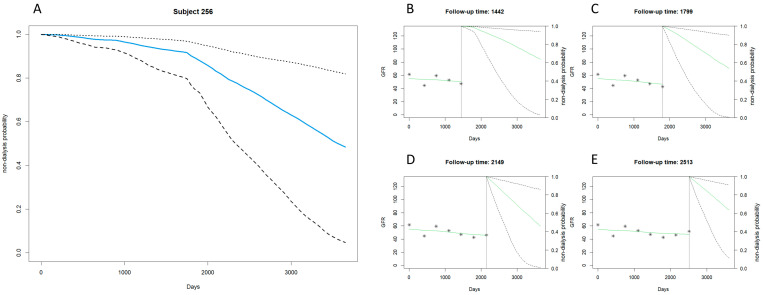
Stage 3a CKD example: late dialysis with the joint model. (**A**) Displays days since observation starts on the horizontal axis (0 to over 3000 days) and the probability of not undergoing dialysis on the vertical axis (0 to 1.0). The solid blue line represents the estimated probability of non-dialysis for case number 256, decreasing over time. Dashed black lines show the 95% confidence interval, widening over time to indicate increasing uncertainty in long-term forecasts. (**B**–**E**) Mark follow-up time at the top, with the solid green line for predicted GFR changes and dashed black lines for the 95% confidence interval of non-dialysis probability, indicating greater uncertainty over time. Asterisks (*) denote actual observed GFR values.

**Figure 2 biomedicines-12-00622-f002:**
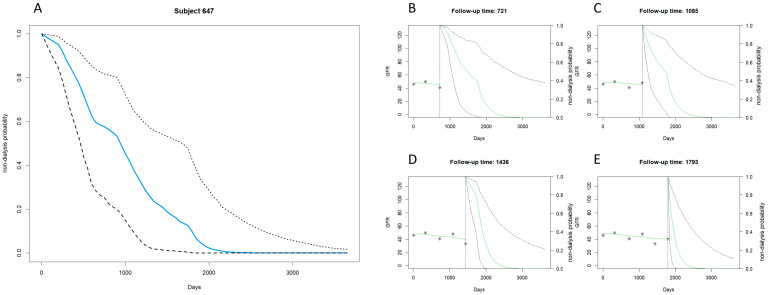
Stage 3a CKD example: early dialysis with the joint model. (**A**) Displays days since observation starts on the horizontal axis (0 to over 3000 days) and the probability of not undergoing dialysis on the vertical axis (0 to 1.0). The solid blue line represents the estimated probability of non-dialysis for case number 647, decreasing over time. Dashed black lines show the 95% confidence interval, widening over time to indicate increasing uncertainty in long-term forecasts. (**B**–**E**) Mark follow-up time at the top, with the solid green line for predicted GFR changes and dashed black lines for the 95% confidence interval of non-dialysis probability, indicating greater uncertainty over time. Asterisks (*) denote actual observed GFR values.

**Figure 3 biomedicines-12-00622-f003:**
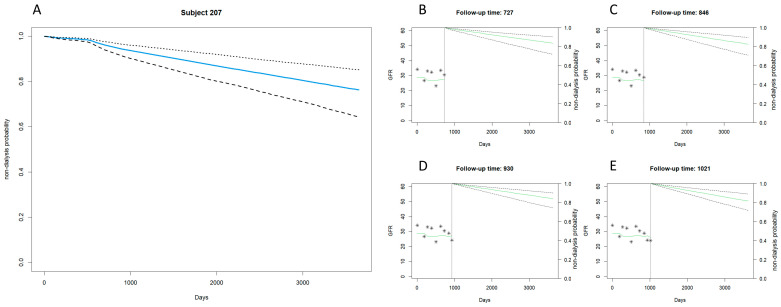
Stage 3b–5 CKD example: late dialysis with the joint model. (**A**) Displays days since observation starts on the horizontal axis (0 to over 3000 days) and the probability of not undergoing dialysis on the vertical axis (0 to 1.0). The solid blue line represents the estimated probability of non-dialysis for case number 207, decreasing over time. Dashed black lines show the 95% confidence interval, widening over time to indicate increasing uncertainty in long-term forecasts. (**B**–**E**) Mark follow-up time at the top, with the solid green line for predicted GFR changes and dashed black lines for the 95% confidence interval of non-dialysis probability, indicating greater uncertainty over time. Asterisks (*) denote actual observed GFR values.

**Figure 4 biomedicines-12-00622-f004:**
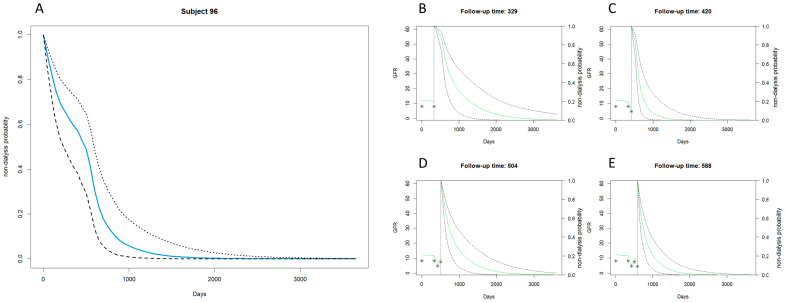
Stage 3b–5 CKD example: early dialysis with the joint model. (**A**) Displays days since observation starts on the horizontal axis (0 to over 3000 days) and the probability of not undergoing dialysis on the vertical axis (0 to 1.0). The solid blue line represents the estimated probability of non-dialysis for case number 96, decreasing over time. Dashed black lines show the 95% confidence interval, widening over time to indicate increasing uncertainty in long-term forecasts. (**B**–**E**) Mark follow-up time at the top, with the solid green line for predicted GFR changes and dashed black lines for the 95% confidence interval of non-dialysis probability, indicating greater uncertainty over time. Asterisks (*) denote actual observed GFR values.

**Table 1 biomedicines-12-00622-t001:** Patient characteristics of group 3a.

Variable	Non-Dialysis	Dialysis	*p*-Value
	Mean (SD)/*n (*%)	Mean (SD)/*n* (%)	
GFR (mL/min/1.73 m^2^)	53.92 (10.21)	46.61 (10.83)	<0.001
Height (cm)	160.52 (11.45)	161.15 (7.89)	0.714
Weight (kg)	65.88 (12.43)	64.59 (12.54)	0.505
SBP (mmHg)	132.86 (17.94)	137.48 (19.00)	0.098
DBP (mmHg)	72.82 (10.90)	71.30 (11.22)	0.371
Cr (mg/dL)	1.30 (0.24)	1.52 (0.42)	<0.001
LDL (mg/dL)	102.39 (28.16)	96.50 (29.71)	0.178
HbA1c (%)	6.52 (1.35)	6.88 (1.72)	0.098
PCRln (mg/g)	5.23 (1.24)	6.11 (1.54)	<0.001
Age (year)	80.74 (11.30)	79.57 (13.09)	0.507
Sex			0.846
Female	144 (30.8%)	13 (28.3%)	
Male	323 (69.2%)	33 (71.7%)	
Hypertension			0.985
Yes	189 (40.5%)	18 (39.1%)	
No	278 (59.5%)	28 (60.9%)	
Diabetes			0.024
Yes	229 (49.0%)	14 (30.4%)	
No	238 (51.0%)	32 (69.6%)	
CVD			0.271
Yes	357 (76.4%)	39 (84.8%)	
No	110 (23.6%)	7 (15.2%)	
Frequency	3.77 (1.70)	3.30 (1.74)	0.080

**Table 2 biomedicines-12-00622-t002:** Joint model results for the group 3a.

Model Component	Parameter	Estimate	Std. Error	*p*-Value
Longitudinal Process	(Intercept)	57.122	3.484	<0.0001
	Obstime	−0.002	0.001	0.0002
	SBP	0.046	0.013	0.0005
	Cr	−2.096	0.210	<0.0001
	Age	−0.088	0.037	0.0167
Event Process	PCRln	0.604	0.146	<0.0001
	Male	0.933	0.382	0.0146
	Diabetes	0.060	0.350	0.8637
	Hypertension	0.531	0.327	0.1042
	Assoct	−0.117	0.027	<0.0001
	log(xi.1)	−9.104	1.824	
	log(xi.2)	−8.044	1.796	
	log(xi.3)	−7.732	1.785	
	log(xi.4)	−8.961	1.827	
	log(xi.5)	−7.576	1.701	
	log(xi.6)	−8.121	1.729	
	log(xi.7)	−6.695	1.664	

**Table 3 biomedicines-12-00622-t003:** Patient characteristics of groups 3b to 5.

Variable	Non-Dialysis	Dialysis	*p*-Value
	Mean (SD)/*n* (%)	Mean (SD)/*n* (%)	
GFR (mL/min/1.73 m^2^)	26.94 (10.74)	11.81 (7.27)	<0.001
SBP (mmHg)	137.72 (19.35)	141.92 (21.98)	0.142
DBP (mmHg)	74.37 (32.27)	75.44 (14.21)	0.802
Hgb (mg/dL)	11.42 (2.05)	9.62 (1.60)	<0.001
Hct (%)	34.63 (5.96)	29.89 (5.33)	<0.001
BUN (mg/dL)	41.44 (22.46)	78.01 (41.92)	<0.001
Cr (mg/dL)	2.68 (1.57)	6.22 (3.40)	<0.001
Na (mg/dL)	140.18 (3.48)	139.77 (5.05)	0.452
Ca (mg/dL)	8.92 (0.60)	8.31 (0.80)	<0.001
P (mg/dL)	3.96 (0.83)	5.06 (1.51)	<0.001
Alb (g/dL)	3.62 (0.46)	3.16 (0.63)	<0.001
Tri (mg/dL)	134.71 (79.79)	159.64 (100.05)	0.039
LDL (mg/dL)	102.32 (33.06)	104.68 (37.91)	0.628
HbA1c (%)	6.50 (1.40)	6.66 (1.37)	0.423
PCRln (mg/g)	6.33 (1.76)	7.69 (1.63)	<0.001
Age (year)	80.25 (13.20)	74.97 (13.55)	0.006
Sex			0.615
Female	88 (32.8%)	22 (37.3%)	
Male	180 (67.2%)	37 (62.7%)	
Hypertension			0.141
Yes	137 (51.1%)	37 (62.7%)	
No	131 (48.9%)	22 (37.3%)	
Diabetes			0.251
Yes	196 (73.1%)	48 (81.4%)	
No	72 (26.9%)	11 (18.6%)	
CVD			0.353
Yes	21 (7.8%)	2 (3.4%)	
No	247 (92.2%)	57 (96.6%)	
Frequency	3.78 (3.20)	4.02 (3.66)	0.611

**Table 4 biomedicines-12-00622-t004:** Joint model results for groups 3b to 5.

Model Component	Parameter	Estimate	Standard Error	*p*-Value
Longitudinal Process	(Intercept)	37.385	<0.001	<0.0001
	obstime	<0.001	0.001	0.4422
	SBP	−0.021	0.001	<0.0001
	DBP	0.007	0.003	0.0083
	Cr	−2.563	<0.001	<0.0001
	P	−0.014	<0.001	<0.0001
	LDL	−0.017	0.001	<0.0001
	PCRln	−0.775	<0.001	<0.0001
	Age	−0.056	0.001	<0.0001
	Hgb	1.049	<0.001	<0.0001
	BUN	−0.043	0.007	<0.0001
	HbA1c	−0.069	<0.001	<0.0001
Event Process	Male	0.339	<0.001	<0.0001
	Diabetes	0.304	<0.001	<0.0001
	Hypertension	0.244	<0.001	<0.0001
	Assoct	−0.153	0.004	<0.0001
	log(xi.1)	−5.964	<0.001	
	log(xi.2)	−6.639	<0.001	
	log(xi.3)	−21.137	<0.001	
	log(xi.4)	−6.201	<0.001	
	log(xi.5)	−4.929	<0.001	
	log(xi.6)	−5.250	<0.001	
	log(xi.7)	−5.553	<0.001	

## Data Availability

The datasets generated and analyzed during the current study are not publicly available due to privacy/ethical restrictions but are available from the corresponding author on reasonable request.

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
