# Peer review of "An Approach for Personalized Dynamic Assessment of Chronic Kidney Disease Progression Using Joint Model"

_biomedicines, 2024, doi:10.3390/biomedicines12030622_

Round 1

Reviewer 1 Report

Comments and Suggestions for Authors

1.      What are the “yes” and “no” in the column of Level in Table 1? Please add the explanation bellowing the table.

2.      How are the p values calculated in Table 1?

3.      Please add the units for each item in Table 1.

4.      For Figures 1-4, please add the different channels like A, B, C etc. for each sub-figure. Also please add the meaning of each line in the figure legend.

5.      What do the asterisks stand for in Figures 1-4? Please add the description in figure legend.

6.      The right channels’ x axis is time in Figures 1-4, please add the unit for the time, like ‘days’.

7.      How to validate the joint model for prediction in CKD assessment? Please add correlated content in the discussion section.

Author Response

We provide a point-by-point response to the reviewer’s comments. Please see the attachment.

Reviewer 2 Report

Comments and Suggestions for Authors

The article by Liao et al. presents an interesting multiparametric assesment od kidney disease progression in a local cohort of patients. The paper is very clear and the study well designed. Results, discussion and limitations are clearly presented. We think the paper can be accepted as it is.

The paper presents a multipapametric model to predict kidney failure in chronic disease patients.

The prediction of disease progression in these cases is very relevant in the clinical managment of these patients.

At present, models based on classical analytical paramenters are unable to succesfully predicts clinical worsening in these cases.

Therefore, a new approach using various of classical parameters combined in a prediction algorith can be helpful.

The multiparametric approach presented is new and apparently effective in predicting clinical worsening in the patient population addressed.

Despite the number of patients is limited, the approach has been succesfully applied.

Limitations have occurred but have been disclosed clearly and discussed by the authors themselves.

Tables, figures and formulas have been clearly presented.

In their present form, Figure 1, 2, 3 and 4 in their right side are too small, but I think these can be improved during the editorial process.

Also, a more detailed explanation in figure legends would greatly help readers.

A noteworthy limitation, albeit anticipated by the authors, is that the small population emploied for validation at a very high age (about 80 years) and in clinical practice the need to predict kidney failure is usually required much earlier in patient's life.

An additional discussion should address how the model could be implemented in clinical centers abroad. Do they considered developing a software package usable in other clinical centers and situations?

Author Response

(The authors gave the same response as above.)

Round 2

Reviewer 1 Report

Comments and Suggestions for Authors

I've no other concerns or comments for revising. The manuscript looks good.